# Using TraDIS as a complementary approach to long term evolution for mapping adaptive mutations in *Escherichia coli*

**Mathew T. Milner**[1,2*], **Hrishiraj Sen**[1], **Manuel Banzhaf**[2], **Peter A. Lund**[1*]

**1** Institute of Microbiology and Infection, School of Biosciences, University of Birmingham, Birmingham, United Kingdom, **2** Biosciences Institute, Faculty of Medical Sciences, Newcastle University, Newcastle Upon Tyne, United Kingdom

\* Mat.Milner@newcastle.ac.uk; P.A.Lund@bham.ac.uk

## Abstract

Long term laboratory-based evolution experiments are a powerful tool that are increasingly being used to study fundamental aspects of evolution and to identify genes that contribute to overall fitness under different conditions. However, even with automation, the time that they take to execute limits the extent to which evolution experiments can be used as part of a high throughput approach to understand the links between genotype and phenotype. Mutations that lead to genetic loss of function (LoF) are frequently selected for in evolution experiments. Thus, in principle these experiments could be done more rapidly by starting not with clonal isolates but with dense transposon libraries that will contain loss of function mutations in all non-essential genes. Here, we test this hypothesis by comparing the results of long term (5 month) evolution experiment, in which *E. coli* was grown with daily transfers in unbuffered LB starting at pH 4.5, with short term (5 and 10 day) experiments on a high-density transposon library in the same strain and under the same conditions. We show that there is a overlap in the genes and pathways identified using the two methods, as well as identifying other gene of interest whose LoF contributes to fitness. This approach has the potential to complement laboratory-based evolution, enabling rapid, higher throughput, testing of a wide range of parameters that may have an influence on evolutionary trajectories.

## Author summary

Understanding how bacterial populations adapt to environmental stress is central to microbiology and evolutionary biology. Laboratory evolution experiments are commonly used to uncover the genetic changes that confer increased fitness, but these experiments are often slow and laborious. In this study, we asked whether the results of long-term adaptive laboratory evolution (ALE) could be

**Data availability statement:** The data that support the findings of this study are publicly available from the European Nucleotide Archive (ENA) under the project accession PRJEB96796.

**Funding:** This work was supported by the Biotechnology and Biological Sciences Research Council via the Midlands Integrative Biosciences Training Partnership Doctoral Training Partnership (MTM), the Leverhulme Trust (Grant RPG-2020-252 to MTM and PL), the Darwin Trust of Edinburgh (HRS), and a UKRI Future Leaders Fellowship (MR/V027204/1 to M.B). The funders had no role in study design, data collection and analysis, decision to publish, or preparation of the manuscript.

**Competing interests:** The authors have declared that no competing interests exist.

predicted using a faster method: short-term selection of a dense transposon mutant library, analysed by transposon-directed insertion site sequencing (TraDIS). We evolved *E. coli* K-12 MG1655 in unbuffered LB at pH 4.5 for 5 months and compared the mutations that arose to those selected after just 10 days of evolution using a high-density transposon library in the same strain and conditions. We observed significant overlap in the genes identified by both approaches, including independent disruptions in shared regulatory pathways. This suggests that short-term selection on a diverse mutant population can uncover many of the same adaptive changes seen in long-term evolution experiments. We also identified and validated the fitness effects of several mutations uniquely found in the transposon approach. Rather than predicting specific mutations from first principles, this method offers a rapid, empirical means of anticipating which genes are likely to be involved in adaptation under defined conditions, helping to guide further mechanistic studies by offering a powerful shortcut for investigating microbial evolution.

## Introduction

Adaptive laboratory evolution (ALE) experiments involving micro-organisms have three main overlapping applications: testing evolutionary theory under controlled conditions, providing insights into connections between genotype and phenotype, and engineering organisms with new and desirable characteristics (for recent reviews on each of these topics see [1–7] respectively). They all use variations on the theme of long-term growth of populations under defined conditions, either in continuous culture, or with repeated serial dilution into fresh media. The advent of cheap whole genome sequencing, coupled with methods for analysing large datasets, has vastly increased our capacity for identifying the mutations responsible for phenotypic changes [8–10]. In addition, automation of repetitive and labour-intensive steps has been used in some cases to increase the throughput of ALE experiments, allowing higher replication and the testing of larger numbers and ranges of potentially selective parameters [11–14]. Thus, ALE has become a key part of the toolkit of many molecular microbiology laboratories.

*Escherichia coli K12,* as an enteric organism and neutrophile, naturally encounters both acidic environments (such as the human gastrointestinal tract) and alkaline environments (for example wastewater or surface waters) [15,16]. Even common laboratory media, if unbuffered, undergo pH shifts during growth. [17] To our knowledge, only a single long-term study has explicitly alternated acidic and alkaline conditions: Hughes *et al.* showed that populations evolved at constant acid or constant alkaline pH developed clear trade-offs, whereas populations evolved under cyclic acid–alkaline regimes became generalists, maintaining high fitness in both environments [18]. Fluctuating pH therefore remains an underexplored evolutionary parameter in *E. coli*, and is an appropriate condition to examine given that real-world settings, are inherently variable rather than constant.

A long-standing question in evolutionary biology, articulated by Stephen Jay Gould [19], is whether evolution would follow the same path if we could "replay the tape of life". Although mutation is random and outcomes of evolution may appear contingent [20–22], adaptive laboratory evolution (ALE) has allowed this idea to be tested directly. Numerous ALE studies have shown that independently evolving populations often converge on similar phenotypes and genetic changes, indicating a degree of predictability despite underlying stochasticity [1,23–25]. The fact that repeated ALE experiments often yield reproducible outcomes shows that evolution under defined conditions is, to some extent, predictable. Prediction of evolutionary trajectories can be approached in two ways: from first principles, which would require a detailed understanding of genotype–phenotype relationships and genotype–environment interactions [26–29], or from precedent, by accumulating enough ALE data to recognise which genes are consistently selected under particular conditions. Over time, insights from repeated ALE experiments can refine genotype–phenotype models, gradually improving our ability to make predictions from first principles.

Several studies using ALE as a tool have shown that loss of function (LoF) mutations are particularly commonly selected for, the term applying to loss of function at the genetic level, for example through partial or complete gene deletion, or the appearance of nonsense mutations [30–36]. This is not universally the case: specific selection pressures (for example, for antibiotic resistance) can favour gain-of-function mutations [37–39]. However, LoF mutations will in general be relatively common, simply on the basis that there will always be many more ways to disrupt the sequence of a gene, and hence the function of the encoded protein, than there are ways to modulate its function in a way that leads to a selective advantage. LoF mutations can result in a selective advantage for the organism carrying them, often (though not invariably) linked to a loss of negative regulation of specific genes or pathways [30,35]. The relatively high frequency of LoF mutations, and the impact of mutations in genes that encode regulators, provides a potential route to dealing with a factor that limits the use of ALE, namely, the time it takes for mutations to arise in the population under study. ALE experiments typically start with a more or less clonal population (e.g., from a culture grown from a single bacterial colony). During the growth of this population, random mutations arise and the organisms carrying those mutations that confer a selective advantage are eventually selected for, and become more ubiquitous in the population. However, the initial stage of waiting for mutations to arise could be circumvented by starting the experiment with a large and diverse transposon library.

Transposon insertions often cause loss of function, meaning that many mutations that will enable a selective advantage under different conditions will already be present in such a library. Moreover, the effects of transposon insertion vary depending on location and/or orientation. Insertions within operons can generate strong polar effects that increase or decrease transcription of downstream genes; insertions near promoters or regulatory regions can alter transcriptional starts sites; and in some cases insertions can demonstrate both transcriptional and translational readthrough leading to expression of neighbouring downstream genes [40–42]. Thus, transposon libraries will be likely to contain mutations causing both loss of function and gain of function that can provide a selective advantage under specific conditions. Because the relative frequency of these mutations in a population can be tracked over time using the sequencing methods developed for Transposon directed insertion-site sequencing (TraDIS), ALE experiments starting with transposon libraries should enable the identification of insertions that increase relative fitness over a shorter timescale than is typical for ALE starting with clonal populations. Such an approach has already been used to identify transposon mutations that may be useful in applied contexts [35,43,44]. Moreover, the mutations identified in this approach should significantly overlap with those identified in conventional ALE experiments. If this hypothesis is correct, ALE on transposon libraries could legitimately be used as a method to address many of the same questions that are studied using ALE experiments on clonal populations. This in turn would enable significant expansion of studies by allowing greater replication, and a broader exploration of multiple selective parameters.

In this study, we therefore tested the hypothesis that evolution of a transposon library over a short time frame can accurately predict the mutations that will arise and be selected for a long term ALE experiment under identical conditions that begins with a clonal population. We also consider the nature of the mutations that arise, recognising that transposon

insertions may not just cause simple loss of function but can also act by causing disruption in the regulation of specific genes in a way that is selectively advantageous under specific conditions.

## Results

### A limited range of mutations are selected for in an environment with fluctuating pH

As part of our studies into acid stress responses in *E. coli*, we investigated the consequences of long-term exposure of *E. coli* K12 MG1655 to an environment where the pH was initially acidic but increased over time, eventually becoming alkaline due to the accumulation of metabolic by-products. Five replicate populations, all derived from the same starter culture, were grown for approximately 5 months with dilution every 24 hours into fresh unbuffered LB at pH 4.5. We refer to this as the Long Evolution Experiment (LEE). During growth, monitoring of culture pH showed very little change for the first two to three hours, followed by a fairly rapid increase from pH 4.5 to pH 7 and then a slower increase to a final pH of approximately 8.8 after 24 hours (Fig 1A). This was likely to be a consequence of the accumulation of waste metabolites from metabolism of peptides, which are the primary energy source in LB [17]. The same pattern of increase in pH was seen if cells were grown in LB starting at pH 7, except that the final pH reached was reproducibly slightly higher (Fig 1B). At the end of this evolution experiment, samples from each population (labelled E1P to E5P) were competed against a Lac- derivative of the starting strain, and in all cases the evolved populations showed a significant improvement in fitness when compared to the parental strain (S1 Fig). Unique clones were isolated from each evolved population and named E1A to E5A, from populations E1P to E5P respectively.

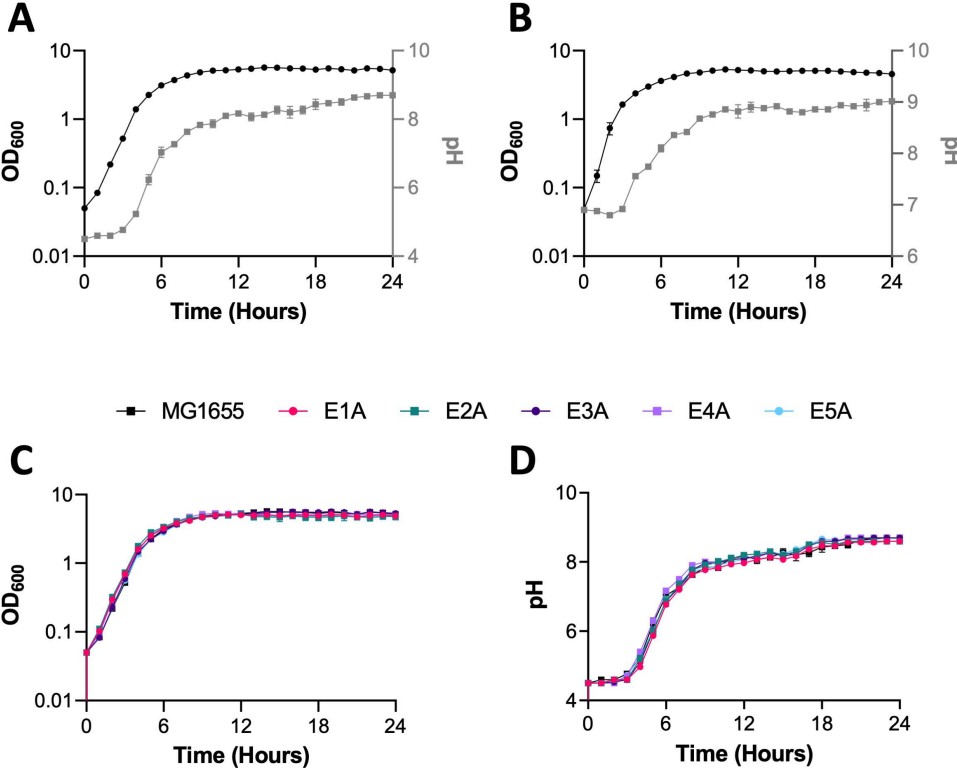

**Fig 1. Growth and pH change in LB, shown for MG1655 and five clonal isolates from evolved MG1655.** Growth (as measured by OD600, shown in black) and pH change (shown in grey) of MG1655 in LB, from an initial pH of 4.5 **(A)** or 7 **(B)**. Comparison of growth **(C)** and pH change **(D)** in LB, initial pH 4.5, of MG1655 and five clonal isolates from five independently evolved populations (E1A to E5A, with colours as shown on figure).

None of strains E1A – E5A showed distinguishable growth characteristics under the conditions of the evolution experiment when compared to the parental strain (Fig 1C and 1D). However, competition experiments showed that all five strains had significantly higher fitness than the parental strain under these same conditions (Fig 2A). To test whether low pH was the major selective parameter, we also competed each of the five strains against the parental strain using LB at an initial pH of 7. Higher fitness was also seen in all evolved strains compared to the parent strain under these conditions (Fig 2B). To identify environmental parameters which were most significant with regard to increases in fitness, we carried out further competition experiments with two of the clonal isolates, E1A and E4A. Initially we tested whether fitness differences were maintained under more aerobic environments, by performing competitions in 50ml LB within 250ml Erlenmeyer flasks. This setup produced comparable fitness estimates within a single day to those obtained from our previous five-day competition assays. Using this faster and more aerated system we then examined how E1A and E4A performed under a range of different growth conditions, namely LB buffered at a starting point of 4.5, 7, or 9, LB with added organic acid (1mM acetic acid), and M9 supplemented with casamino acids and glucose and buffered to a starting pH of 4.5 or 7. The results, summarised in S1 Table, showed that both strains were fitter than the parental strain under all growth conditions in LB, with E4A generally showing the higher fitness. They were also fitter when grown in supplemented M9, though the increases in fitness here were smaller than those seen in LB. It thus appears that the starting pH of the cultures was unlikely to be a significant selective parameter selecting for the fitter genotypes.

We sequenced the genomes of all five clonal isolates E1A to E5A. Novel mutations that are not seen in the parental strains are shown in Table 1. There were clear signs of similar evolutionary trajectories being followed independently in the different strains. All five isolates had new mutations in, or immediately upstream of, *arcA* and *cytR*. Four out of five had an identical mutation in *rpoA*, three out of five had mutations in genes associated with fimbrial production or assembly, and three out of five had mutations in genes associated with tryptophan metabolism. The presence in different strains of identical mutations in *rpoA* raised the possibility of an early selection event for this mutation in one population followed by cross-contamination between the cultures. This was largely ruled out by sequencing the complete genome of 4 clonal isolates taken after 14 days passage from the E1P population, which was found to contain the same *arcA* mutation as E1A but no other mutations; thus, the *rpoA* mutation must have arisen after the *arcA* mutation in this lineage (S2 Table). Many of the mutations seen were point mutations or small indels, but several were also caused by IS insertions, sometimes at

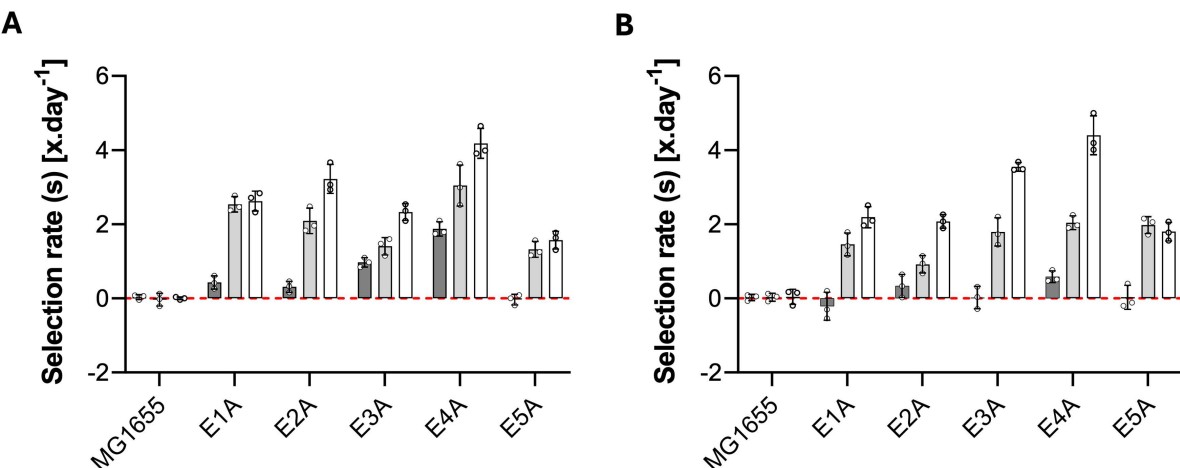

**Fig 2. Relative fitness measured as selection rate (s) of parental MG1655, and evolved clonal isolates, vs KH001 (MG1655 *lac*).** Competition experiments were conducted over 5 days with dilution into fresh media every day. Values of (s) were determined at day 1 (black), day 3 (grey), and day 5 (white), representing the cumulative X·day⁻¹ selection rate for each day with the starting pH being either 4.5 **(A)** or 7 **(B)**. The red dashed line represents no difference in fitness between the competing strains.

**Table 1. Breseq analysis of whole genome sequence data of clonal isolates E1A to E5A.** All comparisons were to the whole genome sequence of the parental MG1655 strain. The occurrence of each mutation in each population is shown as a shaded box if the mutation is present.

| Position | E1A | E2A | E3A | E4A | E5A | Mutation | Annotation | Gene | Description |
|---|---|---|---|---|---|---|---|---|---|
| 1196325 | █ | | | | | Δ 15,088 bp | IS-mediated | [icd]-mcrA | Large deletion: [icd], ymfD, ymfE, lit, intE, xisE, ymfI, ymfJ, cohE, croE, ymfL, ymfM, oweE, aaaE, ymfR, beeE, jayE, ymfQ, stfP, tfaP, tfaE, stfE, pinE, mcrA |
| 1969584 | | | | █ | | Δ 8,919 bp | IS-mediated | [tap]-flhD | Large deletion: [tap]; tar; cheW; cheA; motB; motA; flhC; flhD |
| 3035546 | | █ | | | | T→C | T246A (ACG→GCG) | prfB ← | peptide chain release factor RF-2 |
| 3377183 | | | | █ | | C→T | D80N (GAT→AAT) | sspA ← | stringent starvation protein A |
| 3440150 | █ | | | | █ | T→G | N294H (AAC→CAC) | rpoA ← | RNA polymerase alpha subunit |
| 3544482 | █ | | █ | | | IS186 (-) +5 bp:: _1 bp | coding (359–363/771 nt) | bioH ← | pimeloyl-ACP methyl ester carboxylesterase |
| 3888473 | █ | | | | | C→A | F13L (TTC→TTA) | tnaC → | tryptophase leader peptide |
| 3888626 | | █ | | | | IS5 (-) +4 bp | intergenic (+117/-101) | tnaC →/→tnaA | tryptophase leader peptide/tryptophase/L-cysteine desulfhydrase; PLP-dependent |
| 3888873 | | | | | █ | IS1 (+) +9 bp | coding (144–152/1416 nt) | tnaA → | tryptophase/L-cysteine desulfhydrase; PLP-dependent |
| 4106379 | | | █ | | | G→T | intergenic (+59/-90) | cpxP →/→fieF | inhibitor of the cpx response; periplasmic adaptor protein/ferrous iron and zinc transporter |
| 4123585 | | | | | █ | G→A | P291L (CCG→CTG) | cytR ← | Anti-activator for CytR-CRP nucleoside utilization regulon |
| 4124389 | | █ | | | | 31 bp x 2 | duplication | cytR ← | Anti-activator for CytR-CRP nucleoside utilization |
| 4124521 | █ | | | | | IS5 (+) +4 bp | Intergenic (-65/ +88) | cytR ←/ ← priA | Anti-activator for CytR-CRP nucleoside utilization regulon/Primosome factor n' (replication factor Y) |
| 4124521 | | | | █ | | IS5 (-) +4 bp | intergenic (-65/ +88) | cytR ←/ ← priA | Anti-activator for CytR-CRP nucleoside utilization regulon/Primosome factor n' (replication factor Y) |
| 4360793 | | | | █ | | A→C | L381* (TTA→TGA) | cadC ← | cadBA operon transcription activator |
| 4542161 | | | █ | | | IS5 (-) +4 bp | coding (125–128/597 nt) | fimE → | tyrosine recombise/inversion of on/off regulator of fimA |
| 4545086 | | █ | | | | C→A | intergenic (+57/-10) | fimC →/ → fimD | periplasmic chaperone/fimbrial usher outer membrane porin protein; FimCD chaperone-usher |
| 4639989 | | █ | | | | G→T | N106K (AAC→AAA) | arcA ← | response regulator in two-component regulatory system with ArcB or CpxA |
| 4640014 | | | | | █ | T→G | D98A (GAT→GCT) | arcA ← | response regulator in two-component regulatory system with ArcB or CpxA |
| 4640190 | █ | | | | | C→A | M39I (ATG→ATT) | arcA ← | response regulator in two-component regulatory system with ArcB or CpxA |
| 4640557 | | | | █ | | Δ2 bp:: IS186 (-) +7 bp:: Δ1 bp | Intergenic (+15/-379) | yjjY →/ → yjtD | uncharacterized protein/putative methyltransferase |

identical positions in different strains, though not always with the same orientation. In addition, two clones (E1A and E4A) showed deletions, of approximately 15 kb and 9 kb respectively, both of which were apparently IS-mediated.

We next compared the unique genome sequences of these individual clones with genome sequences from the evolved populations, which we expected to contain different mutations (in addition to those seen in the clonal isolates) at different

frequencies. To do this we sequenced samples from the five evolved populations to approximately 80-fold coverage and analysed the resulting genome sequences of the populations using the *breseq* pipeline with a cut-off of 5% (i.e., if a mutation appeared in less than 5% of the sequencing reads for that population, it was not included in the final list of mutations). The results are shown in S3 Table. The genome sequences of the populations showed, as expected, good agreement with those seen in the clonal isolates that were derived from them. Nearly all the mutations detected in the clonal isolates were also found at high frequencies in the populations from which they were isolated. However, two mutations seen in clonal isolates were absent from the population samples. These were a mutation between *cpxP* and *fieF* that was present in clonal isolate E3A, and a mutation between *fimC* and *fimD* that was present in clonal isolate E2A; neither of these were detected in any population at the cut-off value that we used. In addition, there were two mutations seen in more than one clonal isolate that were only present in a single population sample: a mutation in *bioH* that was present in both E1A and E3A but only seen in E1P, and a mutation in *fimE* that was present in E3A and E4A but only seen in E4P. In no cases were these present at 100% of the population sample, so given the fact that the clonal isolates were sampled randomly from the populations, it seems likely that these simply represent cases where strains with specific mutations were missed by random chance. Many other mutations were observed in the populations, but most of these were present at much lower values than were those that were also present in the clonal isolates. Interestingly, in the populations E3P, E4P and E5P, mutations were also identified in the intergenic region between *gltP* and *yjcO* and were reported to be present in around 10–33% of the population, although these were not seen in their respective clonal isolates. Mutations within this region have previously been identified within a laboratory evolution experiment conducted under the dicarboxylic acid, adipate, potentially suggesting a phenotypic advantage. This observation may be an indication of a subpopulation present within these evolved populations which was not picked up within our clonal isolates. However, this finding was not investigated further in the present study.

## Analysis of an MG1655 TraDIS library after serial passage in LB under fluctuating pH conditions

As discussed in the Introduction, it is reasonable to assume that a significant proportion of mutations found in this type of experiment will result in loss of function at the level of the individual gene, on the basis that there will always be many more ways of removing the function of the encoded protein than changing it to something that is beneficial to the organism in which it is expressed. There is good precedent for this in previous laboratory evolution studies [30,35,45,46]. The fact that many mutations nevertheless reached very high proportions or complete fixation in several discrete evolving populations also strongly suggests that these same mutations caused a gain of function (i.e., an increase in relative fitness under the specific growth conditions used) at the phenotypic level of the cells that carried them, and that there are relatively few ways in which this can happen. This corresponds to the idea in evolutionary theory that there may be a limited set of possible pathways with which to climb a peak in a "fitness landscape" [46–48]. If this is indeed the case, then the mutations that represent these pathways should be easily and quite rapidly identifiable by starting with a random mix of mutations in all non-essential genes and determining how their relative proportions change over a short period of time under a given set of selective conditions, rather than waiting for mutations to arise by random chance. Such populations could be made and examined in a variety of ways. One of these is the construction of a high-density transposon library, followed by its growth under selection, and analysis using methods developed for TraDIS and Tn-seq [49,50]. We therefore set out to directly test the hypothesis that there should be a significant overlap in the genes that show fitness gains when mutated in a TraDIS library, if that library is grown for a short period under the same conditions as those used in the laboratory evolution experiment, and those that are seen in the longer experiment that starts with a clonal isolate.

As there are reports of differences between isolates of the model *E. coli* K-12 strain MG1655 from different laboratories, we constructed a new transposon library in the strain that we had used in the evolution experiment, in order to directly compare its behaviour with that of the parental strain. The library contained >550,000 inserts with a mean spacing of 8.1 bases per insert. Analysis of the library using the log likelihood method described in Goodall *et al.* showed that out

PLOS Genetics

of 4321 annotated genes, 345 were identified as essential, 3675 as non-essential, and 299 as ambiguous [40]. This last category represents genes that cannot be identified as being either essential or non-essential with a high enough degree of statistical support. We compared these genes with those identified in Goodall *et al*. and found a very high degree of similarity (S2 Fig and S4 Table). This was as expected, as although Goodall *et al*. used a transposon library in strain BW25113 for their analysis, this strain is very closely related to MG1655 [51]. The small differences in essential genes that were observed between the libraries in these two strains may be due to differences in density of the library, or differences in the experimental parameters used to construct the library (for instance, the mini-Tn5 used in Goodall *et al*. [40] was marked with a gene for chloramphenicol resistance, whereas the one in this study was marked with one for kanamycin resistance).

To test our hypothesis that short-term evolution of a TraDIS library should produce outcomes that overlap with those seen with long term evolution of the parental strain, we grew three independent populations of the MG1655 TraDIS library in LB at a starting pH of either 4.5 or 7, with serial dilution every 24 hours as done in the original experiment, for ten days. We term this the Transposon Selection Experiment (TSE). All culture volumes and transfer volumes were identical to those used in the initial experiment. Samples were taken after one day, five days and ten days of serial passage for sequencing library preparation as described in Materials and Methods. Following sequence analysis, we used the RNA-seq software, edgeR, as the standard pipeline to derive measurements of relative fitness for each gene. The full dataset is shown in S5 Table.

Two points arose from this preliminary analysis. First, principal component analysis showed that although the replicate populations all started from the same library, they showed increasing divergence over time, with the extent of this divergence being greater in the populations grown at pH 7 (Fig 3A). This effect was also demonstrated by measuring Spearman's correlation coefficient (which uses correlation by rank order rather than absolute value) for all pairwise comparisons of the population samples taken at the same time with the same starting condition. This fell from > 0.9 at day 1 to an average of 0.701 (pH 4.5 population) and 0.441 (pH 7 population) by day ten (S3 and S4 Figs). Thus, although the individual populations are diverging with time, they still show good correlation with each other. Second, we saw a significant drop in

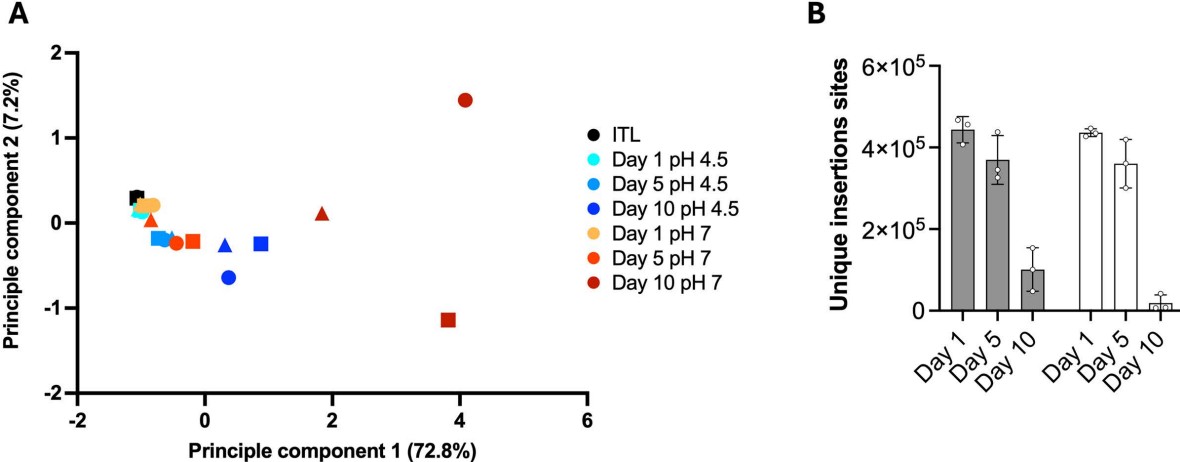

**Fig 3. Changes in the genetic structure of evolving TraDIS libraries over time. (A)** Principal component analysis (PCA) of normalized read counts per gene for each individual sample across the time course of the TSE. ITL represents the initial transposon library, i.e., the original transposon library before any stress is applied, each symbol corresponds to a different replicate of the TSE (circle – S1, square – S2, triangle – S3). **(B)** Number of unique insertions identified in each day for pH 4.5 (Grey bars) and pH 7 (white bars) of the TSE. Individual replicates are plotted alongside the mean and standard deviation.

the number of unique insertion sites in each population over time, corresponding to a progressive loss in complexity of the library (Fig 3B).

This time-dependent loss of complexity within each population suggests that strains carrying many particular insertions become rarer, or are lost completely, during the course of the experiment. This is unsurprising, for two reasons. First, inserts in some genes are expected to cause a loss of fitness when grown in LB, and so these will be depleted and eventually lost from the populations over time, thus reducing overall complexity. Second, even in the unlikely scenario that all inserts were selectively neutral, some stochastic loss of strains containing transposons at different positions will occur each time a serial transfer is completed, although the fact that we did not use a tight bottleneck in our experiments means we would predict the rate of loss via this route to be rather low [52,53]. More importantly, we would expect strains where inserts cause a relative increase in fitness to constitute a progressively larger proportion of the population over time, due to the operation of natural selection. This increase in frequency will inevitably drive down the frequencies of all other strains in the population, even if these have inserts that are initially selectively neutral. From this we can predict that most strains will show an increasing loss of relative fitness over time. To see if this indeed the case, we plotted the ranked relative fitness scores of all genes, averaged across the three populations at pH 4.5 and at pH 7, at day 1, day 5 and day 10. These plots (Fig 4) show that the effect on relative fitness caused by inserts in most genes indeed became more negative over the time of the experiment, while inserts in a small proportion of genes showed a positive impact on relative fitness.

## Genes identified in the TSE and the LEE show significant overlap

The shape of the curves shown in Fig 4 suggested that after 10 days of the TSE, strains with insertions in a small number of genes had accumulated within the population, suggesting that these insertions caused the strains carrying them to show relatively higher fitness than most of the other strains present in the population (see right hand side of graph). This observation was also supported by results presented in S4 Fig, that compares normalised read counts between replicates. However, although our results showed high correlation between replicates (S3 and S4 Figs), the magnitude of the read count corresponding to each specific gene differed. S5 Fig shows the relative frequency of genes (as measured by read count) with insertions that were enriched in the same six evolving populations. As can be seen, certain genes were identified repeatedly in independent populations. Strains carrying insertions in some of these (e.g., *yjjY*, *cspC*, and *yobF*) became enriched in all six populations in the TSE. Others were specific to the initial pH of the population, such as *fimE*

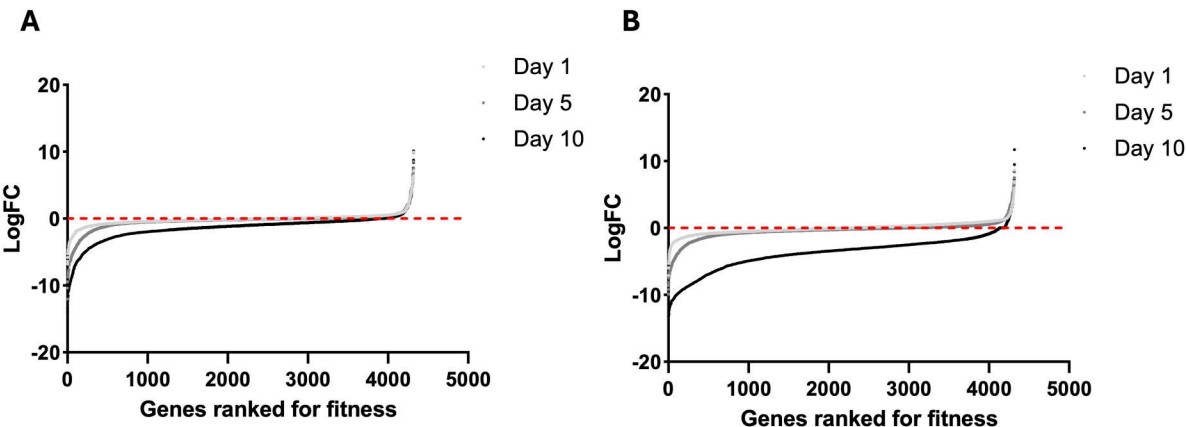

**Fig 4. Relative fitness effect of inserts in all non-essential genes.** Relative fitness was measured by log fold-change relative to insert frequency in the initial TraDIS library. Values are ranked by average log₂ fold change in all conditions. At day 1 (light grey), 5 (grey) and 10 (black) LogFC of all genes were ranked and plotted for each condition, **(A)** Unbuffered LB at pH 4.5 **(B)** Unbuffered LB at pH 7.

and *sspA* (always seen in evolving populations starting at pH 7), and *ptsP*, *cadC*, and *arcB* (always seen in evolving populations starting at pH 4.5). Inserts in these genes were generally distributed randomly and in both orientations, rather than representing enrichment of strains with inserts at one or a few specific positions, consistent with the hypothesis that these inserts caused loss of function at the genetic level (see Figs 7-9). Exceptions to this are considered below where we look at specific examples in more detail.

As our data showed an increase in the frequency of strains with insertions within a few specific genes, we hypothesized that it should be possible to easily isolate strains carrying insertions in these genes from any population after 10 days of the TSE. To test this prediction, a sample from each 10 day population was struck out to single colonies and 10 colonies from each population were screened using PCR for the presence of insertions in either *fimE* or *yjjY*. Of the 60 colonies screened, two strains with insertions in *yjjY* insertions were found (one from Day 10 pH 4.5 and one from Day 10 pH 4.5 S3), and one strain with an insertion in fimE insertion was found, from D10 pH 7 S3. Although not a rigorous analysis, these results strongly suggest that strains with insertions in these genes have accumulated to a high frequency within the populations.

We then tested our hypothesis that the outcomes of the TSE should show significant overlap with those of the Long Evolution Experiment (LEE). Initially, comparisons were made between genes containing insertions that were found to be significantly enriched following analysis using edgeR (logFC > +2/padjust < 0.05) in the evolving TraDIS populations, starting at either pH 4.5 or pH 7, and those that were found to be mutated in at least one of the evolved clonal isolates (Fig 5A) or, more stringently, more than one the evolved clonal isolates (Fig 5B). As can be seen and as noted above, strains with inserts in some genes were enriched in the TSE irrespective of the starting conditions, while others were only enriched in one of the two conditions. Several strains carried insertions in genes that were also mutated in the LEE. Overlaps between the LEE and the TSE were counted if mutations were in the same gene, or in an adjacent gene where the mutation was expected to have a significant effect on the same gene. For example, insertions in *yjjY* (which were only detected in the TSE) almost certainly exert their effect due to changes in *arcA* expression (*arcA* was mutated in the LEE), as discussed further below. Other examples of possible functional relatedness are: *cytR* (found in the TSE starting at pH

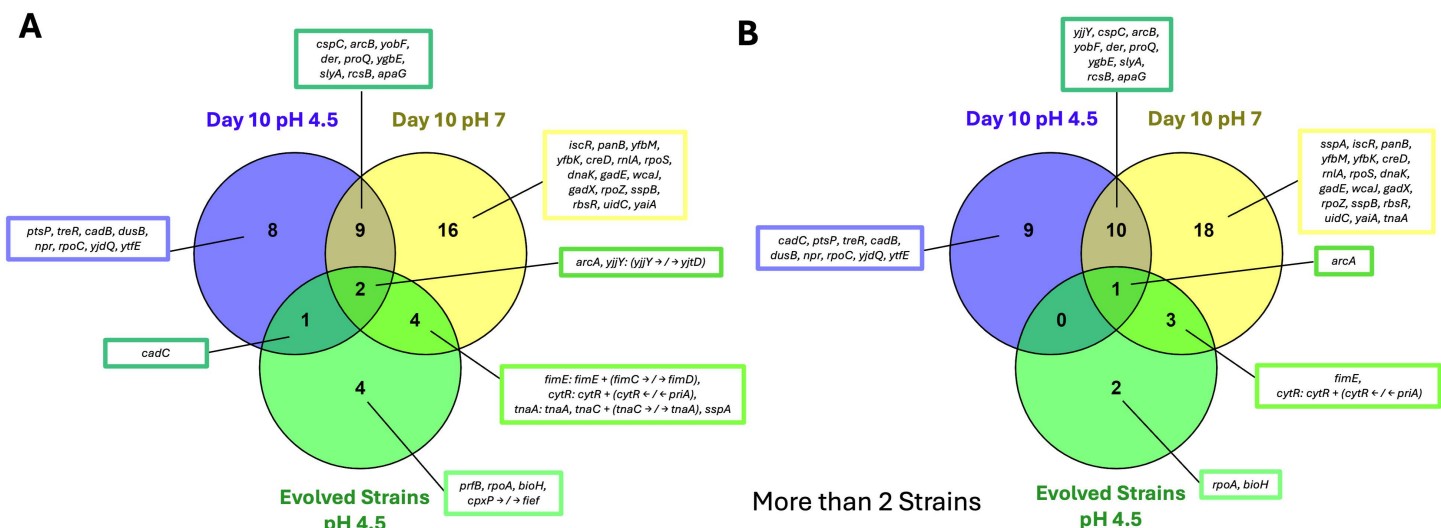

**Fig 5. Comparison of genes identified the TLE with those found in clonal isolates of the LEE.** TLE results are shown at Day 10 pH 4.5 (blue) and Day 10 pH 7 (yellow); LEE results for clonal isolates are in green. **(A)** Mutations which arose in any individual strain **(B)** Mutations which arose in two or more strains. Comparisons were included where TSE-identified genes could be functionally linked to genes or intergenic mutations detected in the LEE, so the overlap reflects the number of LEE gene or intergenic hits associated with TSE genes.

7, while in the LEE mutations/insertions were found in and upstream of *cytR*); *cadC* (found in both the LEE and in the TSE starting at pH 4.5); *cadB* (found in the latter case only); *tnaA* (found in the LEE experiments and the TSE starting at pH 7); *tnaC* (found in the LEE only) (Fig 5B). Of the six genes identified in the LEE that met the more stringent criteria shown in Fig 5B, four were also enriched in TraDIS populations under at least one of the starting conditions. Moreover, of the remaining two, one (*rpoA*) is an essential gene and so would not be represented in the TraDIS data. The second (*bioH*) does show a small but not significant increase in relative fitness in the TSE at pH 4.5 but not pH 7. Possible reasons why it was not seen to be significant in the TSE are considered in the Discussion.

We next compared the data from the TSE with the outcomes in the populations (as opposed to the clonal isolates) of the LEE, with the results shown in Fig 6. This figure shows cases where genes identified in the LEE were also found to be mutated in one or more of the evolving populations at a frequency of at least 5% (Fig 6A) or, more stringently, at least 75% (Fig 6B). A more extensive analysis with intermediate percentages is shown in S6 Fig. These data again support the hypothesis that the TSE can identify genes that, when mutated, are associated with a higher relative fitness in an evolution experiment.

To determine the significance of the degrees of overlap seen, we performed hypergeometric distribution tests using phyper to compare each condition of the TSE and both conditions combined (pH 4.5 and pH 7 gene lists combined) to our LEE results. The results for each comparison are presented in S6 Table. Overall, with exception of one comparison (pH 4.5 pool compared to the LEE), all overlaps were highly significant. We conclude from this that a short time-frame TSE experiment does indeed allow informed predictions about the genetic changes expected in a standard laboratory evolution experiment. Exceptions to this (genes identified using one method but not the other) are considered further below and in the Discussion.

## Investigation of specific examples demonstrates indirect links between insertions and fitness

Within the TSE experiment, several examples were seen of enrichment of strains with inserts in specific genes that were not detected, even at population level, in the LEE. To see whether loss of function of these genes did indeed lead to

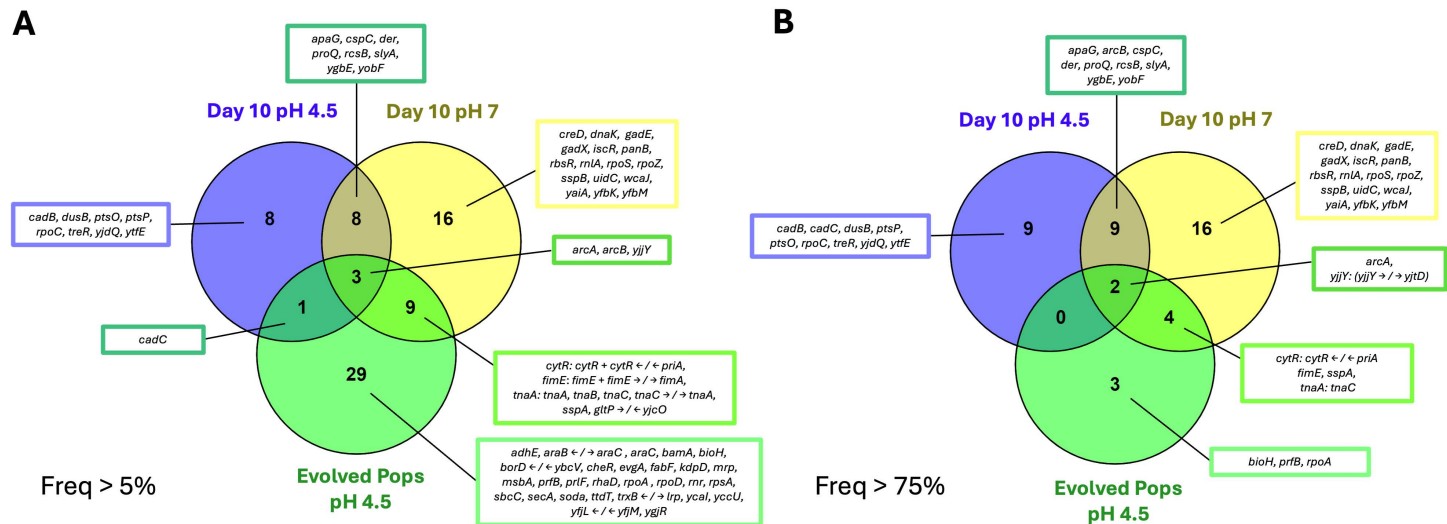

**Fig 6. Comparison of genes identified the TLE with those found in populations of the LEE.** TLE results are shown at Day 10 pH 4.5 (blue) and Day 10 pH 7 (yellow); LEE results for clonal isolates are in green. Frequency cut-offs for the populations were >5% (A) or >75% (B) in at least one population. Comparisons were included where TSE-identified genes could be functionally linked to genes or intergenic mutations detected in the LEE, so the overlap reflects the number of LEE gene or intergenic hits associated with TSE genes.

improved fitness of the strain that contained it, we constructed strains with individual deletion mutations in seven of these genes (all identified only in the pH 4.5 component of the TSE) and measured their fitness relative to KH001 in five-day competition experiments with serial dilution every 24h (Fig 7). With one exception (*cadB*, which is considered in more detail below), all strains containing these knockouts were indeed fitter than the parental strain on day 3 and day 5. Most of them showed a small loss of fitness after 24h, but these were not significant by t-test. Possible reasons for the fact that mutations in these genes were not seen in the LEE are considered in the Discussion.

Further inspection of the insertions in the *cad* operon showed that a large proportion of inserts selected for in the TSE in *cadB* were in an anti-sense orientation relative to the gene for the regulator *cadC* (S7 Fig). Tn5 derivatives can show transcriptional and translational read-through in one orientation due to expression of the antibiotic resistance cassette [40]. We therefore hypothesised that readthrough in the antisense direction of the *cad* operon could, by their effect on *cadC* expression, be reducing or preventing functional expression of *cadA* as well as disrupting *cadB*. To test whether loss of *cadA* affected fitness, we deleted *cadA* and competed the resulting strain with KH001 in the same conditions as used in the 5-day competition experiment. The results show that deletion of *cadA* conferred a fitness advantage. Thus the explanation for the accumulation of inserts in one orientation in *cadB* may be their indirect effect on *cadA* expression, with the gain of fitness caused by the loss of *cadA* expression being sufficient to explain the selective advantage for strains carrying these inserts. This is consistent with previous studies that showed loss of CadA function due to IS-mediated inactivation of *cadC* to be selected for in evolution experiments done at low pH (4.8 – 4.6) [54,55].

Another example where the presence of insertions had an effect on strain fitness that was more subtle than that caused by simple loss of function was in the case of *yjjY* and *arcA*. The *yjjY* gene is immediately upstream of *arcA*, and is annotated as encoding a small (47 amino-acid) protein, possibly associated with the membrane but of unknown function. The adjacent *arcA* gene encodes a global regulator with many diverse targets, acting chiefly under micro-aerobic and anaerobic conditions, and it is subject to complex regulation. In both starting conditions of the TSE we found insertions in *yjjY* to be highly enriched in all day ten populations, with it always being one of the top 10 genes in which insertions were most frequently found (Figs 8A and S5 Fig). Although insertions in *yjjY* in the initial transposon library are distributed evenly in both orientations, the insertions

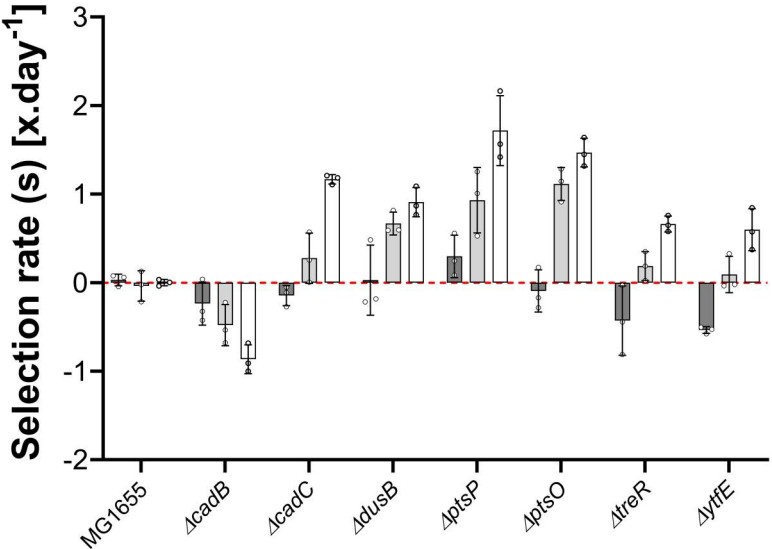

**Fig 7. Competition experiments show that deletion of specific genes enriched in the TSE leads to increased relative fitness.** Selection rates were measured after 1, 3 or 5 days of competition with KH001. Values of (s) were determined at day 1 (black), day 3 (grey), and day 5 (white), representing the cumulative X·day$^{-1}$ selection rate for each day. Individual data points are shown. Bars represent the mean and error bars, standard deviation.

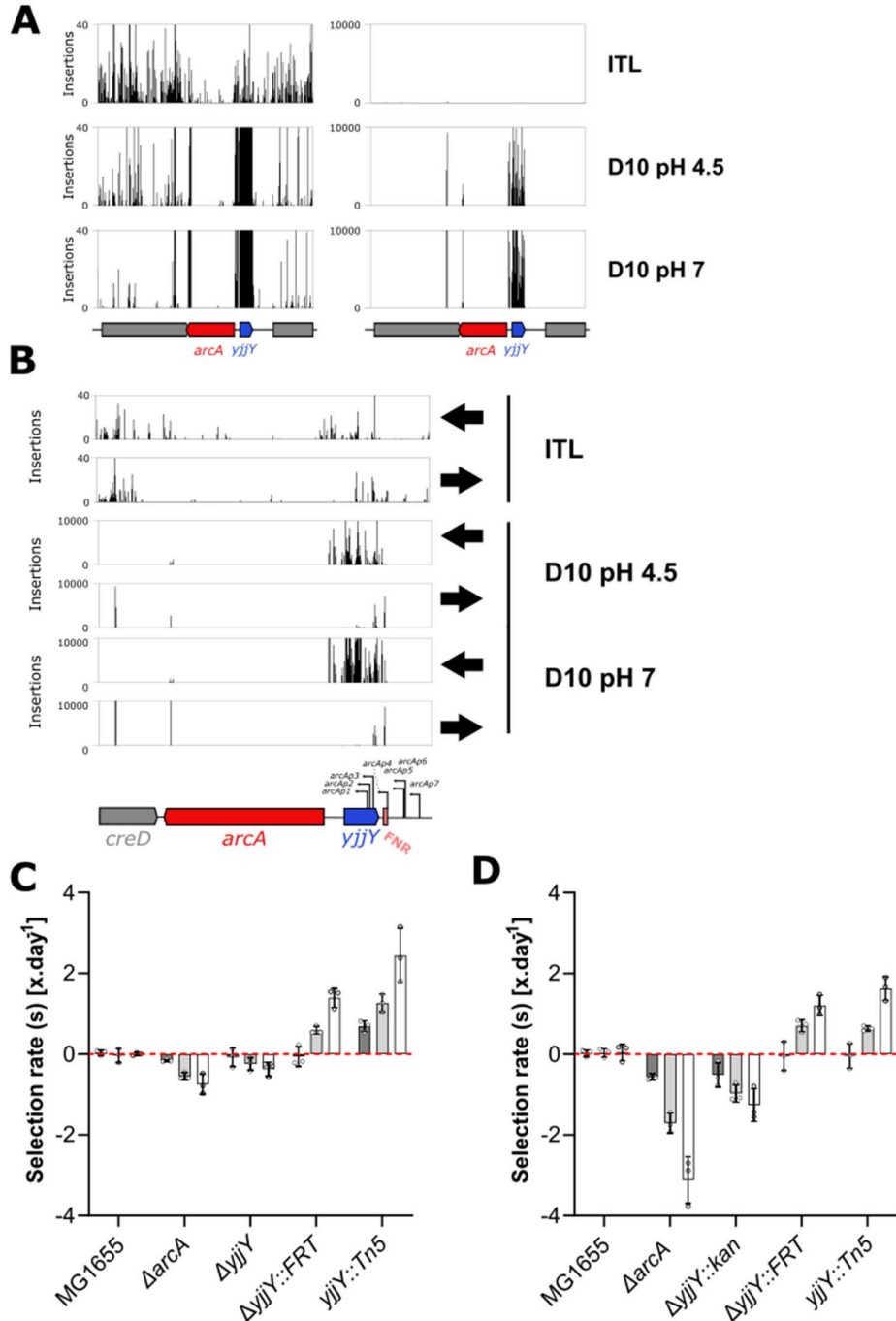

**Fig 8. Impacts of transposon inserts in and close to *arcA*. (A)** Insertion plots showing enrichments of strains carrying inserts adjacent to *arcA* during the TSE. Panels are plotted on different scales, to indicate the extent of enrichments at this site. **(B)** Insertion plots for both orientations of insertion, with the black arrow indicating orientation relative to the orientation of the kanamycin cassette present within the transposon. The diagram shows sites involved in regulation of *arcA* Results of competition experiments looking at fitness of relevant mutations and deletions at starting pH 4.5 **(C)** and pH 7 **(D)**. Selection rates were measured after 1, 3 or 5 days of competition with KH001. Values of (s) were determined at day 1 (black), day 3 (grey), and day 5 (white), representing the cumulative X·day$^{-1}$ selection rate for each day. Bars show the mean and error bars show standard deviation.

that accumulate in the TSE by day 10 are mostly in the sense direction relative to the adjacent *arcA* gene, especially those where cultures were grown starting at pH 7 (Fig 8B). Strains with insertions in *arcA* are rare in the initial library, suggesting that disruption of *arcA* leads to reduction of fitness under the conditions of library construction, so it appears likely that disruption of *arcA* regulation, as opposed to function, may confer some fitness advantage under our conditions.

To test this, we first constructed strains with deletions of either *arcA* and *yjjY* and competed these for five days against KH001 under the same conditions as the TSE. The results of these competitions revealed that strains lacking either *arcA* or *yjjY* had a relative fitness disadvantage under TSE conditions, and this was more prominent at pH 7 than at pH 4.5 (Fig 8C and 8D). To see whether disruption of normal *arcA* regulation conferred a fitness advantage, we isolated a single strain with a Tn5 insertion upstream of *arcA* and in the sense orientation, and competed this against KH001 under TSE conditions. This strain did indeed show a relative fitness advantage after 5 days, with an effect that was more pronounced at pH 4.5 than at pH 7 (Fig 8C and 8D). To see whether this effect could also be caused by disrupting the upstream regulatory region of *arcA*, we removed the kanamycin cassette from a Δ*yjjY* knockout strain from the Keio library, leaving a 82–85 bp scar [56]. This strain (Δ*yjjY::FRT)* also showed an increase in relative fitness under these conditions (Fig 8A and 8C). Overall, these results are consistent with a fitness phenotype for strains carrying inserts in *yjjY* that is caused by altered regulation of *arcA* expression, rather than loss of *yjjY* function per se.

A final example of an indirect effect on fitness revealed by the TSE is given by the *fimE* gene. In the pH 7 condition of the TSE, but not the pH 4.5 condition, strains carrying insertions in *fimE* became very highly enriched (S5 Table and Fig 9A). FimE is a recombinase that, together with *fimB*, regulates expression of the type 1 fimbrae by inverting *fimS,* a regulator region carrying a promoter that in one orientation controls expression of the *fim* operon. *fimS* thus has an ON and an OFF orientation, determining whether or not the *fim* operon is expressed. While *fimE* regulates type 1 fimbrae by switching *fimS* from the ON to OFF orientation, *fimB* is able to switch in both directions (ON to OFF and OFF to ON) creating phase variation within the population. Loss of function of FimE is therefore expected to lead over time to a greater proportion of cells with *fimS* in the ON position. There was no directional bias seen in the *fimE* insertions that accumulated in the TSE (Fig 9B). Insertions in *fimB* were not enriched. This suggested to us that loss of function of *fimE* would lead to a fitness advantage under the TSE pH 7 conditions, and this was confirmed by deleting *fimE* and competing the resulting strain against KH001 for 5 days under pH 7 conditions (Fig 9C). We also showed that deletion of *fimA*, which encodes the major sub-unit of type 1 fimbrae, leads to a relative loss of fitness under the TSE pH 7 conditions, consistent with the hypothesis that the fitness advantage of the *fimE* mutants arises from higher *fimA* expression in these strains (Fig 9C).

To determine whether insertions within *fimE* were specifically associated with growth at pH 7 rather than pH 4.5, we performed a competition experiment under pH 4.5 conditions. Unexpectedly, this revealed that a *fimE* deletion also confirmed a fitness advantage at pH 4.5 as well (Fig 9D). A result which does not explain why strains with inserts in *fimE* are enriched only when grown from a starting pH of 7. This could be explained in terms of the fitness of strains containing different inserts relative to each other, rather than relative to the wild type parent, under different conditions. To see if this was a feasible explanation, we compared the fitness of strains lacking *fimE* (already shown to be fitter than the wild type parent) to the strains described above that carry a Tn5 insert in *yjjY* in the sense orientation relative to *arcA*. It was indeed the case that the strain with loss of *fimE* was less fit than this strain at a starting pH of 4.5, but fitter at a starting pH of 7 (Fig 9E). Interestingly, this was not seen with the Δ*yjjY::FRT* strain described above, which was always less fit than the strain carrying the *fimE* mutation. It can be predicted that the outcome of the TSE will be contingent on the relative fitness of the multiple strains present in the evolving populations, and this experiment shows how relatively slight variations in relative fitness under different conditions may result in significant differences in outcome in quite complicated ways.

## Discussion

It is well established that mutations in certain genes can increase fitness under specific conditions, and transposon-based approaches have repeatedly revealed such adaptive targets. For example, studies in *Salmonella enterica serovar Typhi*

**Fig 9. Effects on fitness in the TSE of mutations affecting type I fimbriae. (A)** Frequencies of strains carrying insertions in the *fim* genes in the initial transposon library (ITL) and after 10 days in the TSE. Panels are plotted on different scales, to indicate the extent of enrichments at this site. **(B)** Insertion plots showing orientation of insertion: black arrow indicates orientation, relative to the orientation of the kanamycin cassette present within the transposon. **(C)** Relative fitness of strains mutated in *fimE* or *fimA* compared with KH001 at pH7; MG1655 is included as a control. **(D)** Competition of *fimE* against KH001 at pH7 and pH 4.5. **(E)** To directly assess the fitness of *fimE* and *yjjY*. A Δ*fimE* mutant was then created in KH001 and competed against *fimE* and *yjjY* mutants in MG1655 at pH 7 and pH 4.5. Selection rates were measured after 1, 3 or 5 days of competition with KH001. Values of (s) were determined at day 1 (black), day 3 (grey), and day 5 (white), representing the cumulative X·day⁻¹ selection rate for each day. Individual data points are shown. Bars represent the mean and error bars, standard deviation.

and *E. coli* identified gene disruptions that enhanced growth under selected environmental constraints [35,57], while recent work in *Klebsiella pneumoniae* demonstrated improved growth in human urine following disruption of regulators such as *hns* [58]. However, to our knowledge, the current study is the first to directly compare the outcomes of evolution from a transposon mutant library with those arising from evolution of a clonal population under identical conditions.

The primary aim of this study was to determine the extent to which short-term laboratory evolution of a high-density transposon mutant library can reproduce the outcomes of longer-term evolution starting from a clonal population. We observed a clear and highly significant overlap between the two approaches, with mutations in many of the same genes conferring a fitness advantage in a 10 day TSE and a 5-month clonal evolution experiment. However, this overlap was incomplete, and each method revealed additional adaptive mutations that were not detected by the other. Notably, the TSE identified a distinct set of genes whose altered function enhanced fitness but which were not selected during the longer-term evolution. We believe this shows that short term evolution studies on transposon libraries have the potential not only to complement and guide longer term ALE experiments, but in some cases to replace long term experiments where the primary goal is the rapid identification of fitness-enhancing mutations across multiple environmental conditions.

Although key genes associated with substantial fitness gains were detected by both approaches, many adaptive loci were uniquely identified by the TSE. For genes identified only in the TSE, we confirmed their association with a fitness advantage by constructing deletions in a subset of these loci and competing them under experimental conditions (Fig 7). This highlights the strength of the TSE in revealing adaptive genes that are not detected in long term ALE experiments. Several explanations may account for why these loci were absent from our LEE. First, some mutations provide benefits only during specific stages of the growth cycle but may be selected against at others, reducing their likelihood of being detected in a typical evolution experiment. This observation is consistent with our findings that several deletions were slightly detrimental after 24 hours yet advantageous after prolonged incubation (Fig 7). Second, mutations which fix early in the LEE can reshape the fitness landscape through epistasis, masking the effects of further beneficial changes that are only detectable in a wild-type background. Third, mutations with weaker selective effects are lost through clonal interference within the LEE, even though the same loci repeatedly demonstrate a fitness advantage within the TSE. Finally, it could also be the limited number of replicate populations in the LEE inherently reduces the likelihood of discovering rarer adaptive targets, whereas the TSE samples a far larger mutational space from the start.

As well as the TSE identifying genes that the LEE missed, the reverse was also true: some genes found in the LEE were not identified in the TSE. Three were identified when a high level of stringency was applied to the data analysis: *rpoA*, *prfB*, and *bioH*. RpoA and PrfB are both essential for growth in LB [40], and so their absence from the TSE dataset is expected: transposon insertions in these genes would be lethal. More significantly, in both cases only a single amino-acid change was selected for, so these likely represent genuine gain of function mutations that could not be considered within a transposon library. Although mutations in *rpoA* are relatively frequently selected for in lab-based evolution experiments, the specific mutation that we observed has not, as far as we can discover, been reported previously in such experiments. The T246A mutation that we observed in *prfB* is known to repair a defect in this protein (a peptide chain release factor; [59]), and a search of the ALEdb database shows it has been reported in other evolution experiments [25]. The third gene, *bioH*, is of interest because in this case the mutation in the LEE was itself caused by an IS insertion, coupled with a single base deletion, in the region upstream of the gene, and very similar mutations would also be expected to be present in the transposon library. Insertions in the *bioH* gene did cause a small but non-significant improvement in fitness at day 10 of the pH 4.5 experiments. We speculate that the impact of mutations in *bioH* may be contingent on the presence of mutations in other genes, but we have not to date studied this example further.

In addition, it is important to note that TraDIS is also well established for identifying genes whose function is required or whose loss of function confers a fitness disadvantage under specific stress conditions. These genes are characterised by a depletion or absence of transposon insertions following growth of libraries under the particular conditions, and are typically referred to as "conditionally essential" genes [40,49,57,58]. We identified these genes as well (S7 Table). There was

no overlap between these genes and those identified within the LEE. This is not unexpected, as disruption or alteration of function of these genes will confer a fitness defect, so strains containing inserts in these genes would be selected against in the LEE. We did not consider this set any further in the current study. Nonetheless, identifying these genes highlights the value of our approach in identifying functions that are critical under a given condition, thereby contributing to understanding of the underlying mechanism of a given stress.

The selective condition we used (growth in unbuffered LB that begins at pH 4.5 and naturally becomes more alkaline) was chosen for its simplicity and relevance to acid adaptation, yet the overlap between our TSE and LEE results indicates that starting pH was not the dominant driver of selection. Most genes in which transposon insertions caused an increase in fitness at pH 4.5 were also identified at pH 7, and examination of the underlying genotypes supports this interpretation. The only clear acid-linked signal was *cadC*, a regulator of the AR4 lysine decarboxylase system previously selected in low-pH evolution experiments [54,60]; in contrast, the opposing fitness effects of mutations in *cadA* and *cadB* suggest that partial disruption of the operon may be beneficial under conditions of fluctuating pH, rather than constant acid stress. Mutations associated with *arcA* (*arcA*, *arcB*, *yjjY*) also matched those observed in rich-medium evolution experiment, and we suggest these mutations are associated with metabolic adjustments that improve fitness under certain conditions [61,62]. Interestingly an LB-based evolution experiment exploring oxygen limitation found mutations in *arcA* and *arcB* were only consistently identified under aerobic conditions, suggesting these mutations may not be linked to oxygen limiting conditions [63]. Similarly, our evolved strains displayed their largest fitness advantages under more aerated conditions, a pattern that may suggest a role for *arcA*-pathway mutations in adaptation to oxygen-rich environments (Fig 2 and S1 Table). The largest overlap observed was between the pH 7 TSE and the LEE conditions, and included genes such as *sspA* and *cytR*. These genes have previously been identified in evolution experiments associated with nutrient scavenging and long-term stationary-phase adaptation [64] and *sspA* has also been implicated in adaptation at pH 9 [65]. Finally, we observed *fimE* mutations in both the LEE and the pH 7 TSE, consistent with selection for a shift toward the fimbrial ON state being advantageous under these conditions. Insertions and mutations within *fimE* have been associated with a wide range of LB-based conditions, including benzoate adaptation, triclosan tolerance, and biofilm-associated fitness [45,66,67]. Overall, these patterns indicate that although low pH exerted a detectable effect on a restricted subset of genes, most adaptive changes reflect responses to general features of the growth regime (particularly prolonged stationary phase, microaerobic conditions, and complex-medium physiology) rather than specific adaptation to acidic conditions.

We observed in our TSE experiments a larger decline in the number of unique insertion sites after 10 days at pH 7 than at pH 4.5 (Fig 4). The reason for this finding is not clear. It could be that the selective pressure is greater in the pH 7 population, although this seems unlikely. It could also be a methodological effect that is sometimes seen in RNA sequencing studies where the presence of highly expressed transcripts can limit the detection of lower abundance transcripts [68]. In our case, strong enrichment of insertions in a small number of genes where mutations cause a large increase in fitness might obscure rarer insertions even if they remain in the population. We have not investigated this further.

In many transposon studies, the fitness advantages associated with individual mutations can be readily explained by simple loss of gene function caused by transposon insertion, which is supported by the observation that deletion of the same genes often yields similar phenotypic gains. However, this is not always the case, as polar effects arising from orientation-dependent transcriptional readthrough from the insert, which can initiate downstream translation, thereby altering expression of neighbouring genes [40–42]. For example, Gray *et al.* showed that insertions of EZ-Tn5 (the transposon present in our library) in *gnd* conferred a fitness benefit not through loss of *gnd* function, but by altering the expression of downstream genes, and this effect was orientation specific. In our work, we also observed several cases which are best explained if the adaptive benefit caused by transposon insertion arises from altered gene expression rather than gene disruption per se [58]. These included (i) modified transcription within the *cad* operon due to antisense transcription from the transposon, (ii) altered regulation of the global regulator (ArcA), and (iii) a change in the frequency of phase variation controlled by *fimE*. In each case, we hypothesise that the regulatory consequences of transposon insertion increased the fitness of the mutant under the selective conditions tested by indirect effects caused by transcription from within the transposon.

These examples show that it is incorrect to assume that transposon insertions act solely by inactivation of gene function, and highlight the importance of examining TSE data in detail, particularly with regard to biased patterns of insertion sites or orientations, which may indicate regulatory effects rather than gene disruption. Exploiting these effects further, for example, through the use of transposons carrying inducible outward facing promoters [68], could provide a means to dissect regulatory networks as well as loss-of-function effects.

Despite its advantages, the TSE approach is inherently limited in several ways. First, it cannot detect adaptive gain-of-function mutations arising from subtle changes in protein sequence. Second, unlike LEE studies, which typically reveal the stepwise accumulation of multiple mutations with additive or synergistic effects, TSE predominantly identifies mutations that confer a benefit in isolation. In principle, epistatic interactions could be probed by generating new transposon libraries in strains containing mutations identified in an initial TSE, but this would add considerably to the experimental burden.

Overall, the overlap between the TSE and LEE, together with the additional set of adaptive genes identified only by the TSE, shows that this approach can address many questions traditionally studied using long-term laboratory evolution experiments, either complementing or replacing such experiments. Such questions may concern the fundamental process of evolution, or the identification of key genes and pathways that determine fitness under specific selective conditions. For example, a central evolutionary question is how selection acts under a defined set of conditions, and how evolutionary outcomes change as those conditions are altered. Since we found that changes in mutant abundance within the TSE library were highly reproducible across replicates (mirroring the repeated selection of similar evolutionary trajectories in long-term evolution experiments) it should be possible to use short-term, high-throughput TSE assays to identify all genes and pathways that respond to selection as particular environmental parameters are varied. Moreover, by systematically altering single selective parameters or combinations of parameters, the TSE could eventually be used to predict which genes and pathways are likely to confer fitness advantages under conditions that have not yet been tested experimentally. These predictions could then be validated empirically, thereby bringing us closer to the ultimate aim of forecasting evolutionary outcomes.

## Materials and methods

### Media and growth conditions

Bacteria were cultured in 5ml volume in 30ml universal vessels at 37°C with shaking (180rpm). Lysogeny broth (LB) was 10g/L tryptone, 5g/L yeast extract and 10g/L NaCl, pH 7. M9 media (42.3 mM $Na_2HPO_4$, 22.1mM $KH_2PO_4$, 8.56 mM NaCl, 18.7mM $NH_4Cl$, 0.1mM CaCl, 2mM $MgSO_4$, pH 7) was supplemented with glucose (0.2% *w/v*) and cas-amino acids (0.2% w/v). For the construction of the transposon library, 2xTY broth was used (16g/L tryptone, 10g/L yeast extract, 5g/L NaCl, pH 6.8). Recovery of transformants was performed in Super Optimum Media (SOC) (20g/L tryptone, 5g/L yeast extract, 10mM NaCl, 10mM $MgCl_2$, 2.5mM KCl, pH 7). For competition experiments, strains were plated onto MacConkey agar supplemented with lactose (40g/L Difco MacConkey Base, 10g/L lactose, pH 7). If required, antibiotics were used at concentrations of 100µg/ml ampicillin, 30µg/ml chloramphenicol and 50µg/L kanamycin.

Media pH was measured using a Thermo Scientific Orion 815600 ROS combination probe and adjusted as required using 1M NaOH and 1M HCl solutions. If required, the following buffers at 50mM concentration were added to maintain the pH: 2-(N-morpholino)ethanesulfonic acid (MES, pKa(37°C) = 5.97); 3-(N-Morpholino)propanesulfonic acid (MOPS, pKa(37°C) = 7.02); N-(1,1-Dimethyl-2-hydroxyethyl)-3-amino-2-hydroxypropanesulfonic acid (AMPSO, pKa(37°C) = 9.10); Homopiperazine-1,4-bis(2-ethanesulfonic acid) (HOMOPIPES, pKa(37°C) = 4.55).

### Strain construction

*E. coli K-12* MG1655 strain was the ancestor of all evolution experiment populations and was the host for the transposon library constructed in the study. Whole genome re-sequencing of this strain revealed an additional IS1 insertion present in the gene *yeaJ* when compared to the reference genome of *E. coli* K-12 MG1655 (Accession: U00096.3). In frame

deletions were constructed using P1 transduction according to the method outlined in [69], transducing marked deletions from the KeiO collection into our strain of interest [70]. Where required the kanamycin resistant cassette was removed using pCP20 [56]. To confirm the deletions PCR was performed using primers flanking the gene of interest.

## Evolution experiment parameters

Starting cultures consisting of 5ml unbuffered LB were inoculated to achieve a starting $OD_{600}$ of 0.05. These were then grown for 24 hours at 37°C with shaking. After 24 hours the culture was pelleted by centrifugation (4000 x g, 10 min), and resuspended in 2ml fresh unbuffered LB. 100µl of this concentrated culture was then diluted into 4.9ml of fresh unbuffered LB, that was then grown for 24 hours before repeating the cycle.

For the laboratory evolution experiment this process of daily passage was repeated for 150 days, using unbuffered LB initially set at pH 4.5. Initially, overnight cultures inoculated from single colonies of *E. coli* K-12 MG1655 were used to start the independent populations. A fossil record of each independent population was created by making a glycerol stock after every 15 days of serial passaging. For the transposon outgrowth experiments, the process of daily passage was conducted for 10 days using unbuffered LB either initially set at pH 4.5 or pH 7. Populations were inoculated directly with a *E. coli* K-12 MG1655 transposon library. A fossil record was created by making glycerol stocks every day of each independent population.

## Genome sequencing

All whole genome resequencing was performed by MicrobesNG (Birmingham, UK) to a minimum of 30X depth for clonal isolates and 80X depth for population samples. Variant calling was conducted using the whole genome resequencing pipeline *breseq* (version 0.37.0) using its default settings [71].

## Transposon library construction

To construct the *E. coli* K-12 MG1655 transposon library, an overnight culture of *E. coli* K-12 MG1655 was diluted into 800ml 2xTY broth at a ratio of 1:114 and grown to reach an $OD_{600}$ of ~0.4 at 37°C with shaking. Cultures were held on ice for 30 mins, pelleted by centrifugation (5000 x *g*, 4°C, 10 mins), the supernatant was removed, and the pellet was resuspended gently in 10% glycerol using half the volume of the original culture. Centrifugation and wash steps were performed repeatedly, reducing the volume by half each time, until a finial volume of 1ml was achieved. This was then divided into 60µl aliquots for transformation. Each aliquot was mixed with 0.2µl of EZ-Tn5<KAN-2>Tnp Transposome (Luigen), transferred to 2mm electroporation cuvette, and electroporated at 2.4kV. Recovery was performed by adding 1ml of SOC media followed by incubation at 37°C for 2 hours. Each transformation was then diluted to obtain single colonies on an LB plate supplemented with 30µg/ml kanamycin. After incubation overnight at 37°C, colonies were then scraped into LB broth. A total of ~1,000,000 colonies were scraped, collected, and pooled, adding glycerol to obtain a 15% *(w/v)* final concentration. The pooled transposon library was then stored at -80°C for future use.

## TraDIS

TraDIS was performed according to Goodall *et al.* [40] Briefly, genomic DNA was extracted from samples using a Stratec RTP Bacteria DNA Mini kit, following protocol 2. DNA was quantified using a Qubit dsDNA HS Assay kit (Invitrogen). 1µg of DNA was then used to prepare a sequencing library. DNA was fragmented using a Diagnode bioruptor plus to achieve an average fragment size of ~350 bp. These fragments were then prepared for sequencing using a NEB Next ultra I kit. Fragments underwent blunt end repair and adaptor ligation according to manufacturer's instructions. Fragment size selection was conducted using AMPure beads. An additional PCR step was added using primers corresponding to the 3' end of the transposon and the universal adaptor sequence to enrich for gDNA fragments containing transposon before a finial PCR step to include illumina flow cell adaptor sequences and a custom inline barcode.

## Data analysis

Data analysis and essential gene prediction was performed according to Goodall *et al*. (2018) [40]. In addition to this, read count data as well as insertion index data was collected for each gene. To compare differences between conditions, the edgeR package (version: 3.26.4) was utilized within R using read counts for each gene [72]. In this analysis two conditions, pH 7 and pH 4.5, were each compared against the ITL. A false discovery rate using a Benjamini Hochberg procedure was applied with a final cutoff value adjusted.pval < 0.05 and a LogFC +/- 2 being used to determine significant difference.

Comparison between gene lists and differences were conducted using Venny 2.0. Hypergeometric tests were performed using a phyper test in R with p value adjusted using a Benjamini Hochberg correction with the p-adjust function. Genome insertion histograms were visualised using the genome browser Artemis.

## Competition experiments

Competition between two strains was done using the *lac⁻* phenotype as the distinguishing marker on MacConkey lactose agar. In the majority of cases KH001 (a *lac* derivatve of *E. coli* K-12 MG1655) was used. Overnight cultures of each strain were mixed into fresh media to achieve an $OD_{600}$ of 0.025 for each strain. This culture was plated onto MacConkey agar and the CFU/ml for each strain calculated to obtain a value at time 0. The culture was grown for a fixed time period t, then plated onto MacConkey agar to obtain the CFU/ml at time t for each strain. Selection rate (equation 1) was chosen as a measure of relative fitness (see S1 Text for more information).

$$S = \ln(\frac{A_t}{A_0}) \, X.day^{-1} - \ln(\frac{B_t}{B_0}) \, X.day^{-1}$$

$$(1)$$

**Equation 1: Selection rate:** The selection rate (S) was calculated from the Malthusian parameters of strains A and B, using their CFU $ml^{-1}$ at time 0 and time t. Here, S denotes the selection rate, while A and B refer to the CFU $ml^{-1}$ of each strain at the two time points. X represents the number of days over which the assay was conducted and is used to express the rate on an $X \cdot day^{-1}$ basis.

Because our competition assays span different durations (1, 3, and 5 days), we chose to report a cumulative selection rate ($X \cdot day^{-1}$) rather than a time-normalised value. Dividing this quantity by the number of days would only rescale its magnitude and would not change which strain is fitter. We therefore treat the cumulative selection rate between strain A and strain B as the most direct summary of competitive performance. This measure remains interpretable when strains increase or decline in abundance and avoids imposing assumptions about how selection acts across the assay period.

## Supporting information

**S1 Text. Overview of relative fitness metrics for competition assays.**
(DOCX)

**S1 Data. Raw data and calculations used in this study.**
(XLSX)

**S1 Fig. Relative fitness measured as selection rate (s) of parental MG1655, and evolved populations, vs KH001 (MG1655 *lac*).** Competition experiments were conducted over a single day in 50ml LB in a 250ml Erlenmeyer flask. The red line indicates expected selection rate, where no difference in fitness was observed.
(TIFF)

**S2 Fig. Comparison of essential genes in BW25113 and MG1655 using TraDIS analysis.** The essential gene list for BW25113 was taken from Goodall *et al*. (2018), while the list for MG1655 was generated from this study. For both strains, essential genes were predicted using insertion index scores modeled with exponential and gamma distributions to represent essential and nonessential gene modes, respectively. Genes were classified as essential, nonessential, or unclear based on a log likelihood score threshold of $\log_2(12)$, corresponding to a 12-fold difference in likelihood between the two modes. Only essential genes were included in this comparison. The percentages in the Venn diagram represent the proportion of unique genes assigned to each group.
(TIFF)

**S3 Fig. Reproducibility of insertion index scores in the LTE.** Each replicate was compared against the other under the pH 4.5 (A-C) and pH 7 (D-F) conditions. Each time point is represented separately: Day 1 (A and D), Day 5 (B and E), Day 10 (C and F).
(PDF)

**S4 Fig. Reproducibility of Read count in the LTE.** Each replicate was compared against the other under the pH 4.5 (A-C) and pH 7 (D-F) conditions. Each time point is represented separately: Day 1 (A and D), Day 5 (B and E), Day 10 (C and F). Not plotted (NP) represents the number of genes not shown due to plotting on logarithmic axis.
(PDF)

**S5 Fig. Proportion of read counts assigned to each gene on Day 10 of the TSE for each replicate (R1–R3).** For each sample, only the top 10 most abundant genes are shown, with the remainder pooled into the "other genes" category. By Day 10, each replicate is predominantly represented by insertions within the top 10 genes only.
(TIFF)

**S6 Fig. Comparison of the Transposon library experiment at Day 10 pH 4.5 (blue), Day 10 pH 7 (yellow) and evolved strains (green).** Different arbitrary frequency cutoffs were used to suggest genes which had undergone stronger selection. Frequency cut offs were as follows (A) No cutoff, (B)>20%, (C) >50%, (D) 90%, (E) Fixation.
(TIFF)

**S7 Fig. *cad* operon results from the TSE.** (A) Insertion plots of the *cad* operon, shown with two different vertical scales to highlight enrichments. Enrichment in insertions is seen in *cadC* and *cadB* on Day 10 for the population with a starting pH pf 4.5. (B) Insertions plotted by orientation of insertion (indicated by the black arrow). (C) Relative fitness of strains carrying *cad* gene deletions at pH 4.5. Selection rates were measured after 1, 3 or 5 days of competition with KH001. Bars represent the mean and error bars, standard deviation.
(TIFF)

**S1 Table. Relative fitness, measured as the selection rate (s), of the parental strain MG1655 and the evolved clonal isolates E1A and E4A, in competition against KH001 (MG1655 Δ*lac*).** Competition experiments were conducted over 24 hours, with time points collected at 6 and 24 hours. Values shown are the mean and standard deviation of three biological replicates.
(XLSX)

**S2 Table. Breseq analysis of whole genome sequence data of 4 Clones isolated from Day 14 of the LEE of E1A.** Clones were labelled F1A-F4A. All comparisons were to the whole genome sequence of the parental MG1655 strain.
(XLSX)

**S3 Table. Breseq analysis of whole genome sequence data of population samples E1P to E5P. All comparisons were to the whole genome sequence of the parental MG1655 strain.** The frequency of each mutation in each population is shown. A mutation was only recorded if it had a frequency of greater than 0.05.
(XLSX)

**S4 Table. Classification of Essential genes in MG1655.** Insertion index scores and log likelihood ratios are shown for MG1655, together with classification for each gene as Essential, Non-essential or Unclear. This data set was compared with a BW25113 data set described in Goodall *et al.* (2018) to generate S2 Fig.
(XLSX)

**S5 Table. Results of the TSE conducted over 10 days with starting pH of 4.5 or 7.** Samples were sequenced on days 1, 5, and 10 and compared to the initial transposon library using edgeR analysis. Three biological replicates were used for each time point. For each comparison, log fold change, log counts per million, p-value, and false discovery rate (FDR) adjusted using the Benjamini-Hochberg correction are shown.
(XLSX)

**S6 Table. Hypergeometric distribution of probabilities comparing the LEE against TSE outcomes.** P-values were adjusted using the Benjamini correction. Clonal comparison refers to the TSE comparison with evolved strains shown in Fig 4. Population comparison refers to the comparison with evolved populations shown in Figs 5 and S6.
(XLSX)

**S7 Table. List of genes classified as fitness-detrimental.** Fitness-detrimental genes were identified using two methods: a log-likelihood approach as described in Goodall *et al.* (2018), and edgeR, based on an FDR < 0.05 and a logFC cut-off < 2. Genes were grouped according to whether they were identified under pH 4.5 only, pH 7 only, or both conditions, for either Day 1 or Day 5 of the TSE. Day 10 was excluded because most of the genes present in MG1655 were classified as having detrimental fitness effects at that time.
(XLSX)

## Acknowledgments

We thank Prof Ian Henderson for providing training and for a helpful discussions that contributed to the development of this work. We are grateful to Dr Emily Goodall and Dr Georgia Isom for their assistance with TraDIS, training, and valuable discussions throughout the project. We also thank MicrobesNG (Birmingham, UK) for their support with whole genome sequencing. We also appreciate the helpful suggestions and feedback by Dr Damon Huber and Dr Jack Bryant.

## Author contributions

**Conceptualization:** Mathew T Milner, Peter A Lund.

**Data curation:** Mathew T Milner, Hrishiraj Sen, Peter A Lund.

**Formal analysis:** Mathew T Milner, Hrishiraj Sen.

**Funding acquisition:** Manuel Banzhaf, Peter A Lund.

**Investigation:** Mathew T Milner.

**Methodology:** Mathew T Milner.

**Project administration:** Peter A Lund.

**Resources:** Manuel Banzhaf.

**Supervision:** Manuel Banzhaf, Peter A Lund.

**Visualization:** Mathew T Milner.

**Writing – original draft:** Mathew T Milner, Peter A Lund.

**Writing – review & editing:** Mathew T Milner, Manuel Banzhaf, Peter A Lund.

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
