## [Decision Letter · Decision Letter 0]

30 Sep 2025

PGENETICS-D-25-00982

Genome-wide screening using TraDIS accelerates identification of key adaptive mutations of longer-term evolution experiment in Escherichia coli.

PLOS Genetics

Dear Dr. Lund,

Thank you for submitting your manuscript to PLOS Genetics. After careful consideration, we feel that it has merit but does not fully meet PLOS Genetics's publication criteria as it currently stands. Therefore, we invite you to submit a revised version of the manuscript that addresses the points raised during the review process.

Please submit your revised manuscript within 60 days Nov 29 2025 11:59PM. If you will need more time than this to complete your revisions, please reply to this message or contact the journal office at plosgenetics@plos.org. Please include the following items when submitting your revised manuscript:

We look forward to receiving your revised manuscript.

Kind regards,

Aretha Fiebig, PhD

Academic Editor

PLOS Genetics

Danielle Garsin

Section Editor

PLOS Genetics

Aimée Dudley

Editor-in-Chief

PLOS Genetics

Anne Goriely

Editor-in-Chief

PLOS Genetics

**Additional Editor Comments:**

Your manuscript has been reviewed by two experts who both agree that the experiments are high quality and valuable to the field. However, reviewer 2 expressed concern that the main conclusion of the manuscript, that transposon mutant pools “accelerate” identification of adaptive mutations, is not well supported by the data. This reviewer has offered several constructive suggestions to address this weakness that involve either reframing of the work and revision of the claims, or additional experimental data. We would be happy to review a revised version of this work.

**Journal Requirements:**

At this stage, the following Authors/Authors require contributions: H Sen. Please ensure that the full contributions of each author are acknowledged in the "Add/Edit/Remove Authors" section of our submission form.

The list of CRediT author contributions may be found here: https://journals.plos.org/plosgenetics/s/authorship#loc-author-contributions

2) We noticed that you used the phrase 'data not shown' in the manuscript. We do not allow these references, as the PLOS data access policy requires that all data be either published with the manuscript or made available in a publicly accessible database. Please amend the supplementary material to include the referenced data or remove the references.

- ® on page: 27

- TM on page: 27.

**Reviewers' comments:**

Reviewer's Responses to Questions

**Comments to the Authors:**

Reviewer #1: The authors describe a comparison of outcomes between a 'conventional' bacterial laboratory evolution experiment, and one seeded with a large transposon mutant library to test the thesis that the mutant library will be able to rapidly predict similar outcomes to the longer conventional experiment. This is tested using E. coli adaptation to media with differing pH levels which is a system the authors have extensive experience with.

I think the paper has been carefully put together and there is a large and thoughtful body of experimental data which is well presented, analysed and discussed. The data clearly shows there is good support for the idea that using TraDIS libraries in experimental evolution can short cut timescales to identifying key genotype-phenotype correlations. This is a useful piece of work which can inspire other experimental design and be a useful reference for the community.

I had a series of relatively minor suggestions:

Line 84 - I'm not sure the labels of 'strong' and 'weak' prediction feel right - particularly as the 'weak' prediction is based on empirical evidence so may often be more likely to be correct?

Line 104 - it's correct that there are more opportunities for loss of function to occur than gain from mutations, but a caveat could be added here that some selective conditions will strongly favour one over the other - e.g. antibiotic exposure experiments typically recover high proportions of missense mutations that alter target sites but are not loss of function per se

Line 230 - I'm not sure its impossible to tell from sequence data what impact a mutation will have and this is often predicted based on prior knowledge

Line 258 - Maybe the library contains >550,000 inserts is safer as the exact number is always a prediction based on the particular experiment and sampling etc

In figure 4 - if I read this right there is generally more loss of mutations over time at ph7 (panel B) than ph 4.5 (A)? IS that not a surprise - wasn't really commented on in the text

Reviewer #2: Summary

The authors use transposon mutagenesis to increase the standing genetic variation that they claim accelerates identification of key adaptive mutations compared to long-term experimental evolution. The main novelty is the combination of a dense transposon libraries and experimental evolution under the same experimental conditions. It is also a strength that identified adaptive mutations are further investigated by pairwise competition experiments with defined mutants. However, there are also substantial weaknesses in that few replicate populations are used and there are few shared mutated genes between experimental evolution lines and Tn libraries. The claim that the approach “accelerates identification of key adaptive mutations” is not well supported and no control experiment where experimental evolution populations are sequenced after a shorter time period than 150 days is included. Despite these weaknesses the study is generally well designed and the manuscript clearly written and could with a reframing of the work and/or additional experimental work be a significant contribution to the field.

Major comments

1. The claim that this approach accelerates “identification of key adaptive mutations” clearly needs a comparison with a short-term experimental evolution setup. Given that only 4-7 mutations/clone are found after 150 days of experimental evolution, many of which are likely loss-of-function mutations, it is possible that the waiting time for mutation will not be limiting (as discussed in lines 112-120) and that adaptive mutations would have been able to increase in frequency enough to be detectable within 10 days. What is needed is deep sequencing of multiple replicate populations at an earlier time point (ideally after 10 days for a direct comparison). The low daily dilution (1:20) in the serial passages means that there are few generations per day requiring a long experiment, but many of the adaptive mutations identified here (gltP, arcA, rpoA, fimE) could be found after a shorter 50-day experiment with a smaller bottleneck (Knöppel et al. 2018 https://pubmed.ncbi.nlm.nih.gov/29755424/ ). If not possible to perform these quite costly additional experiments I suggest refocusing the manuscript to not emphasize that it is faster but rather a complementary approach (see comment 3 below)

2. The introduction focuses to a large degree on loss-of-function mutations which motivates the use of a transposon library that to a large degree is expected to result in disruption of genes. However, the most common mutated shared gene acrA/yjjY does not seems to have loss-of-function mutations. I suggest rewriting the introduction to account for the likely outwards promoter in the transposon and then measure the transcription level of it using qPCR, digital PCR or a reporter gene. This can then be presented as a strength of the method in that mutations increasing expression can be rare in experimental evolution due to a smaller mutational target size. Alternatively, it would be straightforward to confirm changes in transcription for the examples discussed in Lines 564-566). These experiments should be feasible within a couple of months if suitable equipment is available and would significantly strengthen the study.

3. I think the emphasis on the speed of the method compared to regular experimental evolution and finding (very few) shared mutated genes are not the strongest aspects of this major study. Instead, maybe you could consider presenting it as a complementary approach that although it finds shared mutated genes it also has the potential to find others that although they are adaptive might be outcompeted in long-term experiment. It can also be some are not found after experimental evolution because a few mutations of large effects are fixed early on, and epistatic interactions means that mutations that are beneficial in the ancestor are no longer beneficial in the evolved genetic background. Showing that mutations found are adaptive using pairwise competitions (Fig. 7) is exactly the evidence needed to support a claim of being able to find new adaptive mutations not seen after experimental evolution.

4. I think the two pH variations 4.5 and 7 in TraDIS is likely to be confusing for the general reader and might require a better explanation/motivation. Fig. 2C suggest pH is not a major selection pressure, but there seem to be some differences in mutational patterns and surprisingly there are more shared mutations with the pH 7 for the evolved strains (Fig 5, Fig 6. these should also include the pH for the evolved strains). Surprisingly, after the exclusion of pH as a major factor and the identical growth curves (that lack lag phase) of the evolved clones there is little discussion of the identified mutations might increase fitness. For example, arcA is a well-studied global regulator important in microaerophilic environments and involved in regulation of switches between aerobic and anaerobic growth. The small bottlenecks (1:20) means most of the time during the experiments the populations are in stationary phase and a large culture volume in 30ml tubes means poor aeration at high cell densities. It might be interesting to compare the results with Kram et al. 2017 (https://pmc.ncbi.nlm.nih.gov/articles/PMC5340864/) which shows similar mutations (in acrA, cytR, sspA) after long term evolution with prolonged stationary/death phase including long time in stationary/death phase.

5. I am confused by the large differences in selection rates after 1 day in Fig. 2AB vs. 2C. They seem to be much higher in 2C for same strains and same environment, especially for LB pH7 unbuffered. Am I missing something here?

Minor comments

6. I would like to see a convincing motivation for why E. coli K12 in LB is a good model system for this work. To me it seems like K12 would already be highly adapted to a laboratory environment and LB in particular, so that there are few mutations that lead to large increases in fitness. Also, the active IS elements and IS mediated deletions plays a major part in a way that may not be representative for other bacteria and increase the frequency of loss-of-function mutations. Since the experiments is also done at high cell densities could there be issues with the low concentrations of divalent ions that bind LPS and stability the outer membrane.

7. Line 163. “Data not shown.” This data should be added to the supporting information if mentioned.

8. The selection rates in Fig 2, Fig 7 and Fig 9 usually increase or decrease over time. Is this expected and why? In the Travisano Lenski paper cited in the supplementary this is divided with number of days and the equation shown here (line 722) also says day-1, but the calculations are not shown and this kind of increase/decrease is what is expected if not divided by number of days. Also, you do not see any effect on fitness after one day in Fig. 2, can that tell you something about the way that fitness might be increased?

9. In Fig 5., why are there more mutated genes in B than A for the TLE, when B shows those mutated in more than 2 strains? Also in Fig 6, why are there more mutated genes that have >75% freq than >5% for the blue?

10. Line 683. Are they electroporated at 22 kv? Typical for E. coli is around 2.5 kV for a 2 mm cuvette.

11. Lines 684-687. It says each transformation was diluted for single colonies. Then that about 1,000,000 colonies were scraped and collected. Observing single colonies would mean fewer than 1000 colonies per plate, which would mean that >1000 plates were used. It this really the case or am I missing something here?

12. Table 1. at cadC mutation row misspelled “transcriptiol”

13. The manuscript is now quite long and contain 9 figures some of which are very large. I think it can be streamlined and made more accessible by shortening the introduction and possible moving a few figures (or parts of them) to supporting information?

14. Supplementary tables could use a short title in the worksheet explaining what it contains. Source data for and calculations for competition experiments would be useful.

**Have all data underlying the figures and results presented in the manuscript been provided?**

Reviewer #1: Yes

Reviewer #2: **No:** Source data (and ideally calculations) for selection rates for Fig 2AB, 7, 8CD, 9CDE should be added as supporting information. Genome sequencing data and TraDIS data should be deposited in a public repository

PLOS authors have the option to publish the peer review history of their article (what does this mean? ). If published, this will include your full peer review and any attached files.

**Do you want your identity to be public for this peer review?** For information about this choice, including consent withdrawal, please see our Privacy Policy .

Reviewer #1: No

Reviewer #2: No

**Figure resubmission:**
---

## [Decision Letter · Decision Letter 1]

12 Jan 2026

Dear Dr Milner,

We are pleased to inform you that your manuscript entitled "Using TraDIS as a complementary approach to long term evolution for mapping adaptive mutations in Escherichia coli" has been editorially accepted for publication in PLOS Genetics. Congratulations!

Yours sincerely,

Aretha Fiebig, PhD

Academic Editor

PLOS Genetics

Danielle Garsin

Section Editor

PLOS Genetics

Aimée Dudley

Editor-in-Chief

PLOS Genetics

Anne Goriely

Editor-in-Chief

PLOS Genetics

BlueSky: @plos.bsky.social

Comments from the reviewers (if applicable):

Thank you for submitting your work to PLOS Genetics and for your thorough responses to referee feedback. Both reviewers appreciated the revisions and agree that this work is now suitable for publication.

Reviewer's Responses to Questions

**Comments to the Authors:**

Reviewer #1: The authors have carefully revised their manuscript and made sensible changes to introduce some more caveats and highlight some of the unexpected data, i am happy my initial comments have all been addressed now

Reviewer #2: The authors have addressed all comments and made substantial revisions to the manuscript. Specifically the claim that the TSE can accelerate identification of adaptive mutations compared to a conventional evolution experiment has been removed which now means that the conclusions are better supported by the data presented. Although the additional experiments suggested could have strengthened the study they are not required to support the conclusions in the revised manuscript. Also, considering the limited possibilities for the authors to do further experiments I think that the clarifications and revisions are sufficient and I have no further comments.

**Have all data underlying the figures and results presented in the manuscript been provided?**

Reviewer #1: Yes

Reviewer #2: Yes

PLOS authors have the option to publish the peer review history of their article (what does this mean? ). If published, this will include your full peer review and any attached files.

**Do you want your identity to be public for this peer review?** For information about this choice, including consent withdrawal, please see our Privacy Policy .

Reviewer #1: No

Reviewer #2: No

**Data Deposition**

http://datadryad.org/submit?journalID=pgenetics&manu=PGENETICS-D-25-00982R1

**Press Queries**

---

## [Editor Report · Acceptance letter]

PGENETICS-D-25-00982R1

Using TraDIS as a complementary approach to long term evolution for mapping adaptive mutations in Escherichia coli

Dear Dr Milner,

We are pleased to inform you that your manuscript entitled "Using TraDIS as a complementary approach to long term evolution for mapping adaptive mutations in Escherichia coli" has been formally accepted for publication in PLOS Genetics! Your manuscript is now with our production department and you will be notified of the publication date in due course.

With kind regards,

Anita Estes

PLOS Genetics

On behalf of:
